# DYNAMIC $k$-SHOT IN-CONTEXT LEARNING

## ABSTRACT

In-context learning (ICL) allows large language models (LLMs) to learn new tasks from demonstrations and to predict unseen inputs without parameter updates. Existing studies typically fix the number of demonstrations as a static hyperparameter (e.g., 5 or 10), overlooking the variability across models and inputs. We empirically find that the same query text may yield different outcomes depending on the number of demonstrations used. Motivated by this observation, we propose Dynamic-$k$ In-Context Learning (D-$k$-ICL), a novel method that adaptively determines the most suitable number of demonstrations for each query text. The core component is a performance predictor—a neural network that jointly encodes the query text and candidate in-contexts (constructed with varying demonstration counts) to estimate expected inference quality. At inference time, we retrieve the top-$k$ semantically similar demonstrations and progressively vary $k$ to generate candidate in-contexts. The predictor then selects the candidate most likely to yield the best output, thereby dynamically adapting both the number and composition of demonstrations. Across three LLMs and eight datasets, D-$k$-ICL achieves considerable results, with up to 77.8% accuracy, 0.641 MSE, 0.271 ROUGE-1, and 0.295 BLEU. Furthermore, even when trained under few-shot, weakly supervised, or self-supervised settings, the predictor remains effective. Finally, D-$k$-ICL consistently improves performance on commercial LLMs such as GPT-4o, demonstrating its robustness and broad applicability.

## 1 INTRODUCTION

In-context learning (ICL) has emerged as a central paradigm for leveraging large language models (LLMs) to perform downstream tasks without parameter updates (Zhao et al., 2025; Li et al., 2024b; 2025b). While extensive research has examined the selection (Kassianik et al., 2025; Gao et al., 2024; Liu et al., 2024b), formatting (He et al., 2023; Lin & Lee, 2024), and ordering of demonstrations (Oorloff et al., 2025; Xu et al., 2024), relatively little attention has been devoted to the number of demonstrations, commonly denoted as $k$ in $k$-shot ICL. Most existing ICL methods treat $k$ as a fixed hyperparameter, typically determined through heuristics or grid search and applied uniformly across all query texts and LLMs (Mao et al., 2024; Kassianik et al., 2025). In everyday life, it would be unreasonable to expect everyone to wear the same shoe size; instead, shoe sizes should be tailored to each individual's foot length. By analogy, we contend that the number of in-context demonstrations should not be fixed as a static hyperparameter, but rather adaptively chosen based on both the query text and the LLM used during inference. Consequently, fixing the number of demonstrations cannot deliver state-of-the-art performance across all datasets and models.

To test this hypothesis, we conduct comprehensive empirical studies on two representative natural language processing (NLP) tasks (text classification and machine translation) using two recent LLMs (GLM4 9B and Qwen2.5 7B), with the number of demonstrations $k$ varying from $2, 4, 6, 8, 10$. As shown in Fig. 1, our experiments reveal that $k$ has substantial and occasionally non-monotonic effects on performance. Notably, the optimal value of $k$ differs across tasks, models, and even individual test instances, indicating that it should not be treated as a static hyperparameter.

Motivated by these findings, we propose **Dynamic-$k$ In-Context Learning (D-$k$-ICL)**, a novel method that adaptively selects the optimal number of demonstrations for each input. The core idea involves training a performance predictor, which is a neural network that accepts both the query text and a candidate in-context demonstration set with a variable number of examples, and estimates the expected inference performance.

To train the performance predictor for D-$k$-ICL, we construct a dataset of (text, in-context, actual performance) tuples. The labeled retrieval dataset $D_{\text{retrieval}}$ is randomly partitioned into a context retrieval set $D_{\text{context}}$ and a text set $D_{\text{text}}$. For each text $x_{\text{tx}}^i$ in $D_{\text{text}}$, we retrieve the top-$k$ most semantically similar examples from $D_{\text{context}}$, ranked by descending similarity to $x_{\text{tx}}^i$. These demonstrations are assembled into $k$ candidate in-contexts for $x_{\text{tx}}^i$, where the $j$-th candidate in-context ($1 \leq j \leq k$) contains the top-$j$ most similar demonstrations. Each text $x_{\text{tx}}^i$ paired with its $k$ candidate in-contexts is processed by the LLM. The LLM's output is compared with the ground-truth label $y_{\text{tx}}^i$ to compute an evaluation metric (e.g., MSE for regression, BLEU for translation), which defines the actual performance. Finally, we train a dual-input, single-output neural network that takes the text and a candidate context as input to predict the corresponding actual performance.

Similarly, during inference, for each text $x_{\text{test}}^i$ in the test dataset, we retrieve the top-$k$ most semantically similar demonstrations from the retrieval dataset $D_{\text{retrieval}}$ to construct $k$ candidate in-contexts. The $w$-th candidate in-context ($1 \leq w \leq k$) comprises the top-$w$ most similar demonstrations. Each text $x_{\text{test}}^i$ is then paired with its $k$ candidate in-contexts are fed into the trained performance predictor, generating $k$ corresponding performance scores. The in-context yielding the greatest predicted performance is selected for $x_{\text{test}}^i$, with its size determining the optimal number of demonstrations for that $x_{\text{test}}^i$.

We evaluate D-$k$-ICL across five tasks, eight datasets, and three LLMs. D-$k$-ICL achieves considerable results, outperforming the second-best baseline by average margins of 5.67% in accuracy, with corresponding reductions of 0.066 in MSE and improvements of 0.007 in ROUGE-1 and 0.044 in BLEU. D-$k$-ICL also achieves state-of-the-art performance on the proprietary GPT-4o model, attaining 65.8% accuracy. The approach exhibits strong generalization capabilities, transferring effectively both across LLMs and across datasets. Furthermore, D-$k$-ICL functions as a plug-and-play module that enhances the performance of existing ICL methods. Our contributions are summarized as follows:

- We conduct the first systematic empirical study of demonstration number in ICL, revealing its significant but previously underexplored impact.

- We propose D-$k$-ICL, a general and efficient framework that dynamically selects the number of demonstrations via performance prediction.

- We demonstrate that D-$k$-ICL achieves considerable results on five tasks, eight datasets, and three LLMs, and further show that: (i) it generalizes robustly across models, datasets, and tasks; and (ii) it can be used as a plug-and-play module to enhance other ICL methods.

## 2 RELATED WORK

Current research on ICL predominantly addresses three critical dimensions (Mavromatis et al., 2023; Li et al., 2024b; Zhao et al., 2025): demonstration selection, formatting strategies, and optimal ordering (Lin & Lee, 2024; Li et al., 2025b). These factors are systematically utilized to optimize LLM performance (Kassianik et al., 2025).

**Demonstration Selection.** Current approaches fall into unsupervised and supervised paradigms. Unsupervised methods typically retrieve top-$k$ nearest neighbors using similarity metrics (cosine/L2 distance) over embeddings (Tanwar et al., 2023; Qin et al., 2023; Wang et al., 2025), with extensions like $k$NN-based retrieval (Liu et al., 2022a; Cao et al., 2025) and multilingual adaptationsTanwar et al. (2023); Li et al. (2024b). Alternative metrics include mutual information (Sorensen et al., 2022; Zhao et al., 2025), perplexity (Gonen et al., 2023a), and model-generated probabilities (Nguyen & Wong, 2023; Chen et al., 2025; Liu et al., 2024a).

**Demonstration Reformatting.** Reformatting techniques enhance alignment with LLM behavior. Self-generated demonstrations (Kim et al., 2022) synthesize examples without training data, while structured prompting (Hao et al., 2022) modifies attention mechanisms via positional embeddings. Representation-level methods (e.g., ICVs (Liu et al., 2024a), Feature-Adaptive Prompting (Li et al., 2024a)) adapt latent features during inference.

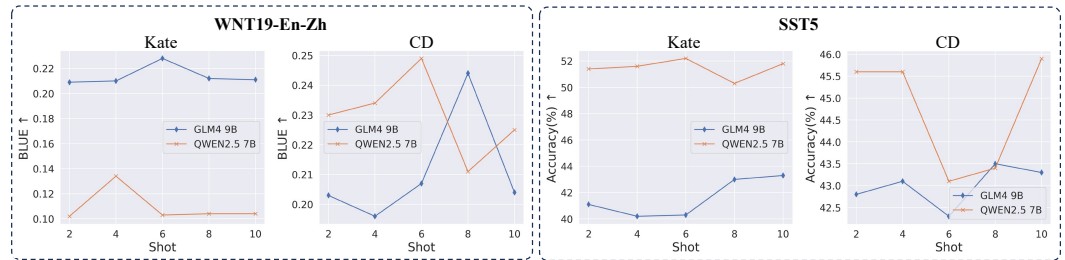

Figure 1: The results of the empirical study conducted on the WNT19-En-Zh and SST5 datasets.

**Demonstration Ordering.** Order sensitivity is well-documented: entropy-based metricsLu et al. (2022), similarity-driven proximityLiu et al. (2022a), and curriculum-based difficulty ranking (ICCL (Liu et al., 2024c)) have been proposed to optimize sequence effects.

Current ICL methods predominantly neglect the influence of demonstration quantity on model performance Kassianik et al. (2025); Zhang et al. (2024). The prevailing approach employs a fixed k-value as a hyperparameter in demonstration selection (Li et al., 2023; Liu et al., 2022a; Rubin et al., 2022; Qin et al., 2023). Nevertheless, existing literature offers insufficient justification for specific k-value choices, especially concerning their adaptability to diverse task scenarios—an area that warrants further investigation.

## 3 EMPIRICAL STUDY

To examine how the number of demonstrations affects ICL performance, we evaluate two ICL methods—KATE Liu et al. (2022b) and Cluster-Diversity (CD) Naik et al. (2023)—on the SST-5 (text classification) and WMT19-En-Zh (machine translation) datasets using GLM4 9B and Qwen2.5 7B LLMs. As illustrated in Figure 1, the inference performance of ICL exhibits considerable variation across datasets and LLMs as the number of demonstrations changes. For example, on the SST5 dataset with the GLM4 9B LLM and Kate ICL method, increasing the number of demonstrations leads to substantial fluctuations in performance, ranging between 40.2% accuracy and 43.3% accuracy. Moreover, we observe that the optimal number of demonstrations differs depending on the dataset, LLM, and ICL method. For instance, on the WNT19-En-Zh dataset using the GLM4 9B LLM and the Kate ICL method, the best performance is achieved when the number of demonstrations is set to 6. In contrast, for the SST5 dataset with the Qwen2.5 7B LLM and the CD ICL method, optimal performance occurs at 10. These empirical results suggest that rather than being a fixed hyperparameter, the number of demonstrations ought to be dynamically determined according to the LLM and input text.

## 4 METHOD

Based on the preceding analysis, we conclude that the number of demonstrations should be dynamic and performance-driven. In particular, the optimal number of demonstrations varies adaptively for each LLMs and each query text $x$. To address this, we propose a **performance predictor** of D-$k$-ICL, denoted as $\mathcal{P}$, which estimates the expected task performance of a given query when paired with the in-context containing a specific number of demonstrations. Formally, given a query $x$ and a candidate in-context $C_x^j$ (comprising $j$ demonstrations), the predictor outputs a score $\mathcal{P}(x, C_x^j)$, which approximates the `actual performance` of $x$ with in-context $C_x^j$.

To train the predictor of D-$k$-ICL, as Fig. **??** shows, we construct a dataset of (`text`, `in-context`, `actual performance`) tuples. This dataset serves as the training input for the **performance predictor** $\mathcal{P}$. After training, for each query text $x_i$ in the test dataset $D_{\text{test}}$, we apply a chosen ICL method to generate multiple candidate in-contexts, each with a different number of demonstrations. The trained **performance predictor** $\mathcal{P}$ is then used to estimate the expected performance for each candidate. The candidate in-context associated with the greatest predicted performance score is selected as the final in-context, along with its corresponding number of

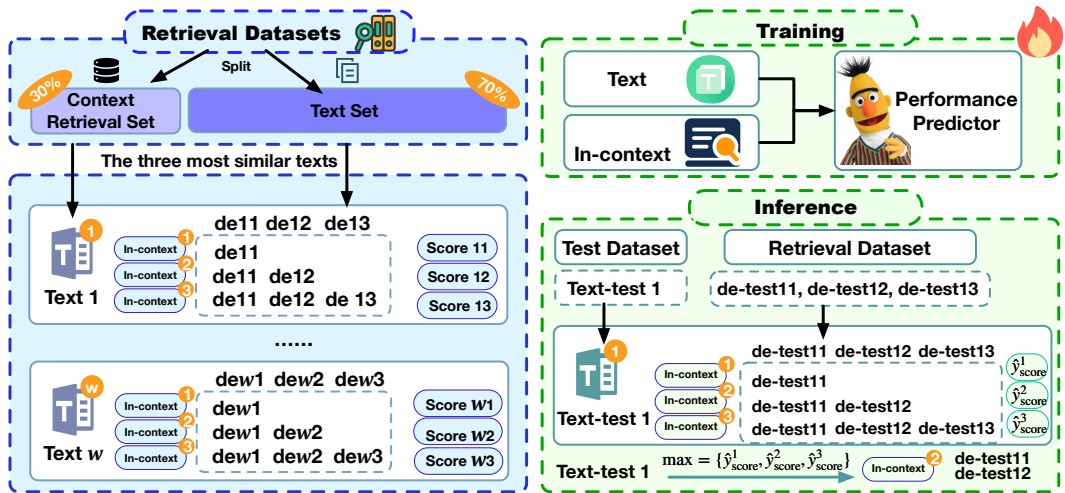

Figure 2: Overall framework of Dynamic-$k$ In-Context Learning (D-$k$-ICL).

demonstrations. Our proposed D-$k$-ICL framework comprises three key stages: ❶ Constructing the training dataset for the **performance predictor** $\mathcal{P}$. ❷ Training the **performance predictor** $\mathcal{P}$. ❸ Using the trained **performance predictor** $\mathcal{P}$ to identify the optimal number of demonstrations and corresponding in-context for each query text $x_i$ in the test dataset $D_{\text{test}}$.

## 4.1 TRAINING DATASET CONSTRUCTION

To train this predictor $\mathcal{P}$, we require a dataset of (text, in-context, actual performance) tuples. In the following, we provide a detailed description of how the text, in-context, and the corresponding actual performance are constructed.

**Training-text construction:** As Fig. 2 shows, the retrieval dataset $D_{\text{retrieval}}$ is randomly partitioned into a context retrieval set and a text set $D_{\text{text}}$. Formally:

$$D_{\text{retrieval}} = D_{\text{context}} \cup D_{\text{text}}, \quad D_{\text{context}} \cap D_{\text{text}} = \varnothing \tag{1}$$

The text $x_{\text{tx}}^i$ in the text set $D_{\text{text}}$ serves as the source of texts used in constructing the (text, in-context, actual performance) tuples.

**Candidate in-contexts construction:** For each text $x_{\text{tx}}^i$ in $D_{\text{text}}$, we select the candidate in-context with different demonstration numbers from the context retrieval set $D_{\text{text}}$. In previous works on ICL, numerous methods have employed similarity metrics to select relevant demonstrations Zhou et al. (2024); Liu et al. (2022b). Building on these methods, we also select the most semantically similar demonstrations from $D_{\text{text}}$ to $x_{\text{tx}}^i$. The process is outlined as follows:

(1) *Text Vectorization*: We begin by vectorizing all texts in $D_{\text{context}}$ and $D_{\text{text}}$ using a pretrained model. Specifically, the vector representations for the texts $x_{\text{tx}}^i \in D_{\text{text}}$ and the context texts $t_{\text{con}}^j \in D_{\text{context}}$ are computed as follows:

$$\mathbf{v}(x_{\text{tx}}^i) = f_{\text{pre}}(x_{\text{tx}}^i), \quad \mathbf{v}(t_{\text{con}}^j) = f_{\text{pre}}(t_{\text{con}}^j), \tag{2}$$

where $f_{\text{pre}}$ denotes the pre-trained encoder.

(2) *Semantic similarity computation*: We compute the cosine similarity between each text vector $\mathbf{v}(x_{\text{tx}}^i)$ and every context vector $\mathbf{v}(t_{\text{con}}^j)$, denoted as $CS_{ij}$. The cosine similarity is defined as:

$$CS_{ij} = \frac{\mathbf{v}(x_{\text{tx}}^i) \cdot \mathbf{v}(t_{\text{con}}^j)}{\|\mathbf{v}(x_{\text{tx}}^i)\|\|\mathbf{v}(t_{\text{con}}^j)\|}, \tag{3}$$

where $\mathbf{v}(x_{\text{tx}}^i)$ and $\mathbf{v}(t_{\text{con}}^j)$ represent the vector representations of $x_{\text{tx}}^i$ and $t_{\text{con}}^j$, respectively. The semantic similarity between $x_{\text{tx}}^i$ and all context samples is given by $\mathbf{S}_i = \{CS_{i1}, CS_{i2}, \cdots, CS_{in}\}$, where $n$ is the total number of $D_{\text{context}}$.

(3) *Selection and Ranking*: After computing the cosine similarities, we select the top $k$ most similar context examples. We denote the $k$ most similar texts to $x_{\text{tx}}^i$ and their corresponding labels, ranked in descending order according to cosine similarity, as follows:

$$\left\{ (t_{x_{\text{tx}}^i}^1, y_{x_{\text{tx}}^i}^1), (t_{x_{\text{tx}}^i}^2, y_{x_{\text{tx}}^i}^2), \cdots, (t_{x_{\text{tx}}^i}^k, y_{x_{\text{tx}}^i}^k) \right\}, \tag{4}$$

where $t_{x_{\text{tx}}^i}^k$ denote the $k$-th most similar text to $x_{\text{tx}}^i$, and let $y_{x_{\text{tx}}^i}^k$ be the corresponding label of that text.

(4) *Construction of in-contexts with varying numbers of demonstrations*: The first candidate in-context for $x_{\text{tx}}^i$ is $C_1(x_{\text{tx}}^i) = \{(t_{x_{\text{tx}}^i}^1, y_{x_{\text{tx}}^i}^1)\}$, the second candidate in-context is $C_2(x_{\text{tx}}^i) = \{(t_{x_{\text{tx}}^i}^1, y_{x_{\text{tx}}^i}^1), (t_{x_{\text{tx}}^i}^2, y_{x_{\text{tx}}^i}^2)\}$, and, in general, the $k$-th candidate in-context is $C_k(x_{\text{tx}}^i) = \{(t_{x_{\text{tx}}^i}^1, y_{x_{\text{tx}}^i}^1), (t_{x_{\text{tx}}^i}^2, y_{x_{\text{tx}}^i}^2), \cdots, (t_{x_{\text{tx}}^i}^k, y_{x_{\text{tx}}^i}^k)\}$.

**Actual Performance Construction:** To illustrate the construction of *Actual Performance*, we take the classification task as an example. Given the text $x_{\text{tx}}^i$ and its associated candidate in-context $C_k(x_{\text{tx}}^i)$, we query the LLM to obtain the predicted label $\hat{y}^{i,j}$. Formally, the prediction is defined as $\hat{y}_{\text{tx}}^{i,j} = f_{\text{LLM}}\left(x_{\text{tx}}^i \mid C_k(x_{\text{tx}}^i)\right)$, where $f_{\text{LLM}}(\cdot \mid \cdot)$ denotes the output of the LLM given the candidate in-context. The *Actual Performance* score is then defined as

$$AP^{i,j} \begin{cases} 1, & \hat{y}_{\text{tx}}^{i,j} = y_{\text{tx}}^i, \\ 0, & \text{otherwise.} \end{cases} \tag{5}$$

For tasks with score outputs, `Actual Performance` is defined as the mean squared error (MSE) between the LLM's predictions and the ground-truth scores. In machine translation, we employ the BLEU score to measure the similarity between generated and reference translations, using this value as the `Actual Performance` metric. For other tasks, we compute appropriate task-specific evaluation metrics by comparing LLM predictions with ground-truths, with the resulting scores serving as the `Actual Performance` measure.

## 4.2 Model Training

When training the performance predictor $\mathcal{P}$, we adopt a dual-input single-output model architecture for the performance predictor $\mathcal{P}$. The two inputs correspond to the query text and the in-context, respectively, while the output represents the `actual performance` score associated with the pair (e.g., a classification label such as 0 or 1, or a BLEU score in translation tasks). The detailed model architecture and training procedure are described in the section titled **Model Architecture and Training Details** (Section 5.1).

## 4.3 Predicting Optimal Demonstration Number

For each text $x_{\text{test}}^i$ in the test dataset, we apply the same retrieval strategy to

$$\left\{ (t_{x_{\text{test}}^i}^1, y_{x_{\text{test}}^i}^1), (t_{x_{\text{test}}^i}^2, y_{x_{\text{test}}^i}^2), \cdots, (t_{x_{\text{test}}^i}^k, y_{x_{\text{test}}^i}^k) \right\}, \tag{6}$$

where $t_{x_{\text{test}}^i}^j$ denotes the $j$-th most semantically similar text to $x_{\text{test}}^i$, with $y_{x_{\text{test}}^i}^j$ being its corresponding label. The tuple $(t_{x_{\text{test}}^i}^j, y_{x_{\text{test}}^i}^j)$ thus represents the $j$-th most similar demonstration.

$$C_j(x_{\text{test}}^i) = \{(t_{x_{\text{test}}^i}^1, y_{x_{\text{test}}^i}^1), (t_{x_{\text{test}}^i}^2, y_{x_{\text{test}}^i}^2), \cdots, (t_{x_{\text{test}}^i}^k, y_{x_{\text{test}}^i}^j)\}. \tag{7}$$

Next, we employ the performance predictor $f_{\text{p}}$ trained in Section 4.2 to estimate the performance score for each candidate in-context . We then select the candidate in-context with the greatest predicted performance score as the final in-context, and use its corresponding demonstration number of in-contexts as the final demonstration number for $x_{\text{test}}^i$.

# 5 Experiments

## 5.1 Experimental Setup

**Tasks and Datasets** To comprehensively evaluate the effectiveness of D-$k$-ICL, We investigate the results of D-$k$-ICL in 5 NLP tasks across 8 widely used benchmark datasets. Specifically: ❶ Machine Translation: machine translation automatically converts text from one language into another.

Table 1: The results of D-$k$-ICL and other methods. Best results in **bold**; second-best underlined.

| LLMs | GLM | Llama | Qwen | GLM | Llama | Qwen | GLM | Llama | Qwen | GLM | Llama | Qwen |
|---|---|---|---|---|---|---|---|---|---|---|---|---|
| Metric | Accuracy (%) ↑ | | | Accuracy (%) ↑ | | | ROUGE-1 ↑ | | | ROUGE-1 ↑ | | |
| Task & Data | Classification: Emotion | | | Classification: SST5 | | | Summarization: Gigaword | | | Text expansion: Gigatiny | | |
| BM25 | 54.3 | 38.0 | 59.0 | 34.2 | 31.4 | 49.9 | 0.024 | 0.020 | 0.108 | 0.122 | 0.128 | 0.135 |
| CD | 16.0 | 39.8 | 55.5 | 43.3 | 36.4 | 45.9 | 0.043 | 0.025 | 0.134 | 0.172 | 0.102 | 0.240 |
| Kate | 63.5 | 40.0 | 72.3 | 43.9 | 35.7 | 51.8 | 0.057 | 0.032 | 0.106 | **0.203** | 0.125 | 0.253 |
| DKNN | 70.5 | 46.5 | 43.0 | 49.8 | 35.7 | 45.0 | 0.033 | 0.028 | 0.114 | 0.167 | 0.107 | 0.237 |
| TTF | 52.5 | 38.5 | 52.9 | 45.0 | 38.5 | 46.8 | 0.043 | 0.036 | 0.114 | 0.163 | 0.099 | 0.264 |
| ICCL | 52.3 | 38.2 | 69.7 | 46.2 | 31.7 | 48.8 | 0.043 | 0.032 | 0.106 | 0.194 | 0.115 | 0.230 |
| PPL | 60.2 | 39.4 | 51.0 | 48.2 | 32.1 | 48.7 | 0.043 | 0.037 | 0.115 | 0.197 | 0.110 | 0.249 |
| D-$k$-ICL | **73.0** | **64.0** | **77.8** | **50.5** | **45.2** | **52.9** | **0.062** | **0.050** | **0.140** | 0.202 | **0.140** | **0.271** |
| Metric | MSE ↓ | | | MSE ↓ | | | BLEU ↑ | | | MSE ↓ | | |
| Task & Data | Textual Similarity: STS14 | | | Textual Similarity: STSB | | | Translation: WNT19-En-Zh | | | TQA: EN-CS | | |
| BM25 | 2.192 | 1.404 | 0.797 | 0.993 | 1.146 | 0.763 | 0.032 | 0.033 | 0.040 | 3488.983 | 531.930 | 352.005 |
| CD | 2.831 | 1.263 | 0.965 | 1.272 | 1.291 | 0.898 | 0.204 | 0.134 | 0.225 | **388.383** | 686.084 | 354.231 |
| Kate | 0.920 | 1.441 | 0.813 | 1.070 | 1.138 | **0.747** | 0.211 | 0.137 | 0.104 | 452.645 | 499.811 | 343.921 |
| DKNN | 1.087 | 1.511 | 0.804 | 1.258 | 1.425 | 0.781 | 0.202 | 0.138 | 0.174 | 453.106 | 517.100 | 354.981 |
| TTF | 3.041 | 1.578 | 0.748 | 1.343 | 1.583 | 0.982 | **0.231** | 0.146 | 0.008 | 498.559 | 666.192 | 401.761 |
| ICCL | 2.252 | 1.361 | 0.727 | 1.118 | 1.199 | 0.786 | 0.214 | 0.139 | 0.111 | 392.296 | 484.441 | 348.676 |
| PPL | 2.538 | 1.372 | 0.989 | 1.101 | 1.403 | 0.816 | 0.219 | 0.144 | 0.165 | 410.250 | 462.403 | 347.495 |
| D-$k$-ICL | **0.827** | **1.241** | **0.641** | **0.790** | **1.128** | 0.767 | 0.222 | **0.219** | **0.295** | 453.131 | **427.227** | **338.486** |

We assess performance on the WNT19-En-Zh (Zhang & Wang, 2025) and WNT19-EN-CS (Novak & Svoboda, 2025) dataset. ❷ Text Expansion: As a generative task, text expansion involves extending a brief input into a more detailed and coherent expression while retaining its original meaning. For evaluation, we use the Gigatiny (Liu & Zhou, 2025) dataset. ❸ Text Summarization: summarization aims to generate concise summaries that preserve the key information of the source text. We conduct experiments on the Gigaword (Napoles & Dredze, 2025) dataset. ❹ Semantic Textual Similarity (STS): This task measures the degree of semantic similarity between sentence pairs. We evaluate performance using the STS-Benchmark (STS-B) Cer et al. (2017) and STS14 (Cer & Diab, 2025) datasets. ❺ Text Classification with SST5 Socher et al. (2013) and Emotion Saravia et al. (2018) dataset. Detailed information regarding the tasks and datasets can be found in Appendix Sections C and D.

**Metrics, baselines and LLMs:** For metrics, we use BLEU (Johnson & Li, 2025) for machine translation, ROUGE-1 (Cheng & Tan, 2025) for text expansion and summarization, MSE (Huang & Zhao, 2025) for STS, and accuracy (Park & Kim, 2025) for text classification. Lower MSE values indicate better performance, while higher BLEU, ROUGE-1, and accuracy scores correspond to stronger results. The formal definitions and computational procedures for these metrics are provided in Appendix B. Meanwhile, we consider TTF (Liu et al., 2025), Delta-KNN (DKNN) (Li et al., 2025a), Clustering-Retrieval (CR) Li & Qiu (2023), KATE Liu et al. (2022b), Cluster-Diversity (CD) Naik et al. (2023), ICCL Liu et al. (2024b), and PPL (Gonen et al., 2023b) as comparative baselines. Moreover, experiments are conducted on three large language models (LLMs): GLM-4 9B Zeng et al. (2024), LLaMA 3.1 8B Touvron et al. (2024), and Qwen2.5 7B Team (2024).

**Model architecture and training details and prompt:** We adopt BERT-base-uncased (Devlin et al., 2019) as the backbone encoder and augment it with five hidden layers. The model is trained for 10 epochs with a learning rate of 1e-3. Optimization is carried out using the AdamW optimizer (Loshchilov & Hutter, 2025) with a linear warmup scheduler to enhance training stability. Additionally, a dropout rate of 0.2 is applied in the classification layers to mitigate overfitting. In addition, the specific prompts for different datasets are provided in Appendix Sections D.

**Other setup:** For the baseline methods, the number of in-context demonstrations is fixed at 10. In D-$k$-ICL, the maximum number of demonstrations is also set to 10 for a fair comparison. The pretrained model used in Equation 2 is `MiniLM-L6-v2`. We randomly partitioned the retrieval dataset into a context retrieval set (30%) and a text set (70%).

## 5.2 MAIN RESULTS

Table 1 compares the performance of D-$k$-ICL against baseline methods. D-$k$-ICL consistently achieves considerable results across diverse tasks. Specifically, it improves the BLEU score by an average of 0.044 in machine translation, and enhances ROUGE-1 by averages of 0.008 and 0.006 for text summarization and expansion, respectively. For STS, it reduces the MSE by an average of

0.066. In classification tasks, D-$k$-ICL outperforms the second-best baseline by an average of 5.67% in accuracy. These results demonstrate that dynamically adjusting the number of demonstrations enables D-$k$-ICL to achieve substantial and consistent performance gains across all evaluated tasks.

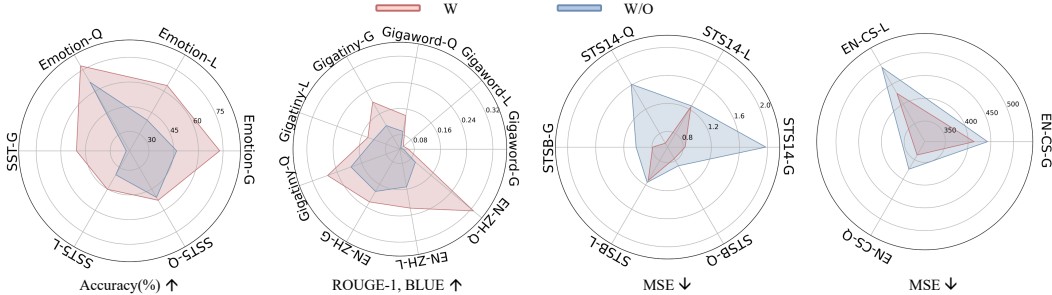

Figure 3: The results of the ablation study comparing D-$k$-ICL with and without the dynamic selection process. D-$k$-ICL incorporating the dynamic selection mechanism achieves significantly superior performance. The notation Emotion-Q refers to the results of the Emotion dataset obtained with the Qwen model; likewise, Emotion-G and Emotion-L refer to those with the GLM and Llama models. This suffix convention (-Q, -G, -L) is consistently used across all datasets.

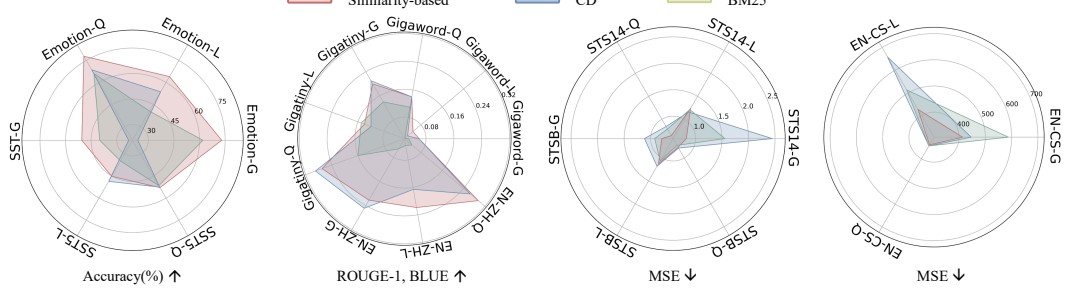

Figure 4: The results comparing performance with and without similarity-based retrieval. D-$k$-ICL augmented with similarity-based retrieval achieves superior performance. The notation Emotion-Q refers to the results of the Emotion dataset obtained with the Qwen model; likewise, Emotion-G and Emotion-L refer to those with the GLM and Llama models. This suffix convention (-Q, -G, -L) is consistently used across all datasets.

## 5.3 ABLATION STUDY

**With and without the dynamic selection process:** For the D-$k$-ICL variant without dynamic selection, the context is constructed by retrieving the top 10 demonstrations using the D-$k$-ICL retriever and directly concatenating them for the LLM. As illustrated in Fig. 3, dynamically determining the number of demonstrations yields superior results. Specifically, for machine translation, it improves BLEU by an average of 0.1 over the strongest static baseline; for text summarization and expansion, it enhances ROUGE-1 by averages of 0.025 and 0.059, respectively; for semantic STS, it lowers MSE by an average of 0.404; and for classification tasks, D-$k$-ICL exceeds the best static baseline by an average of 17.4% in accuracy.

**With and without similarity-based retrieval:** Since D-$k$-ICL utilizes similarity-based retrieval for in-context demonstrations, we investigate the impact of this mechanism by comparing it with alternative retrieval strategies from BM25 and CD. As shown in Fig. 4, the incorporation of similarity-based retrieval consistently improves the performance of SICL. For example, on the Emotion dataset using the Qwen LLM, D-$k$-ICL with similarity-based retrieval achieves an 11.76% higher accuracy than its counterpart without this mechanism.

## 6 ANALYSIS

**Plug-and-play integration for other ICL methods:** D-$k$-ICL can function as a plug-and-play component to enhance existing in-context learning (ICL) methods. As shown in Table 2, integrating D-$k$-ICL's dynamic demonstration selection mechanism with baseline ICL methods yields an average accuracy improvement of 5.01% on the SST-5 and Emotion datasets.

Table 2: The results of plug-and-play integration.

| Method | With SICL | | Without SICL | |
|---|---|---|---|---|
| | Emotion | SST5 | Emotion | SST5 |
| CD | 44.3 | 42.4 | 39.8 | 36.4 |
| Kate | 43.5 | 38.4 | 40.0 | 35.7 |
| ICCL | 44.6 | 36.3 | 38.2 | 31.7 |
| PPL | 45.9 | 38.0 | 39.4 | 32.1 |

Table 3: The performance predictor produced in D-$k$-ICL generalizes across different datasets

| Test Data | WNT19-En-Zh | | | | | |
|---|---|---|---|---|---|---|
| Metric | BLEU ↑ | | | | | |
| LLMs Train Data | GLM | | Llama | | Qwen | |
| | WNT19-En-Zh | STSB | WNT19-En-Zh | STSB | WMT19 | STSB |
| Result | 0.222 | 0.206 | 0.219 | 0.214 | 0.295 | 0.246 |

**Increasing the maximum number of demonstrations enhances performance.** We evaluate D$k$ICL with maximum demonstration numbers set to 4, 7, and 10. As shown in Tab. 4, increasing this maximum consistently enhances model performance across all datasets. Specifically, when the maximum number rises from 4 to 7 to 10, accuracy on the Emotion dataset improves from 58.8% to 61.5% to 64.0%; MSE on STSB refines from 1.165 to 1.134 to 1.128; and ROUGE-1 on the Gigatiny dataset increases from 0.125 to 0.133 to 0.140.

**Application to GPT-4o**: D-$k$-ICL demonstrates strong compatibility with proprietary LLM, including closed-source and commercial APIs such as GPT-4o. As reported in Table 6, D-$k$-ICL achieves SOTA performance on the Emotion dataset using GPT-4o, attaining an accuracy of 65.8%.

**Generalization Ability of the Performance Predictor in D-$k$-ICL:** (1) The performance predictor produced in D-$k$-ICL generalizes across different models. As shown in Tab. 5, when the D-$k$-ICL trained on the Emotion dataset with Qwen and then applied to a test model Llama, it still achieves an accuracy of 57.8%. Although this is lower than the accuracy obtained when training directly on Llama (by 6.2% accuracy), it remains higher than the best accuracy of other ICL baselines (by 11.3% accuracy). (2) The performance predictor of D-$k$-ICL demonstrates strong generalization capability across datasets. As shown in Table 3, D-$k$-ICL—trained solely on the STS-B dataset—achieves a BLEU score of 0.214 on the WMT19-En-Zh machine translation dataset using the Llama LLM, surpassing all baseline methods. This result underscores the exceptional cross-task and cross-dataset generalization ability of the D-$k$-ICL framework.

## 7 DISCUSSION

**Training the performance predictor of D-$k$-ICL on open-source and free LLMs and applying it to commercial models.** D-$k$-ICL requires access to the test LLM during training to construct the `actual performance` values. Consequently, if the test LLM is commercial (e.g., GPT-4o), additional costs are incurred. However, as shown in Section 6, the performance predictor trained with D-$k$-ICL generalizes effectively across models. To reduce cost, we construct `actual performance` values and train the predictor on free, open-source LLMs, before applying it to commercial models. As reported in Tab. 6, when trained on GPT-4 9B with the SST5 dataset, the predictor still achieves an accuracy of 59.2% on GPT-4o, outperforming all baseline methods.

Table 4: Results with different maximum demonstration numbers.

| | Accuracy (%) ↑ | | ROUGE-1 ↑ | | MSE ↓ | | BLEU ↑ | MSE ↓ |
|---|---|---|---|---|---|---|---|---|
| Maximum Number | Emotion | SST5 | Gigaword | Gigatiny | STS14 | STSB | WNT19-En-Zh | EN-CS |
| 10 | 64.0 | 45.2 | 0.050 | 0.140 | 1.241 | 1.128 | 0.219 | 427.227 |
| 7 | 61.5 | 43.8 | 0.048 | 0.133 | 1.249 | 1.134 | 0.205 | 430.740 |
| 4 | 58.8 | 42.1 | 0.045 | 0.125 | 1.287 | 1.165 | 0.190 | 438.791 |

Table 5: The performance predictor produced in D-$k$-ICL generalizes across different models

| Data | | Emotion | | | | | | | |
|---|---|---|---|---|---|---|---|---|---|
| Metric | | Accuracy (%) ↑ | | | | | | | |
| Test model Train model | GLM | GLM Llama | Qwen | GLM | Llama Llama | Qwen | GLM | Qwen Llama | Qwen |
| Result | 73.0 | 66.0 | 72.3 | 55.5 | 64.0 | 57.8 | 74.8 | 71.5 | 77.8 |
| Data | | STSB | | | | | | | |
| Metric | | MSE ↓ | | | | | | | |
| Test model Train model | GLM | GLM Llama | Qwen | GLM | Llama Llama | Qwen | GLM | Qwen Llama | Qwen |
| Result | 0.790 | 0.816 | 0.814 | 1.146 | 1.128 | 1.156 | 0.790 | 0.802 | 0.767 |

Table 6: The results of GPT-4o with SST5 dataset. D-$k$-ICL indicates training on GPT-4o; D-$k$-ICL (GLM) indicates training on GLM and testing on GPT-4o.

| BM25 | CD | Kate | DKNN | TTF | ICCL | PPL | D-$k$-ICL | D-$k$-ICL (GLM) |
|---|---|---|---|---|---|---|---|---|
| 52.5 | 51.8 | 57.3 | 53.1 | 54.5 | 56.4 | 56.9 | 65.8 | 59.2 |

Table 7: Performance of D-$k$-ICL trained on weakly-supervised, unlabeled, and few-shot datasets. The evaluation metric is Accuracy (%) ↑. The numbers 5 and 10 denote the number of demonstrations.

| Shot | 5 | 10 |
|---|---|---|
| BM25 | 42.0 | 45.9 |
| CD | 12.8 | 12.8 |
| Kate | 44.8 | 50.4 |
| DKNN | 13.6 | 11.6 |
| TTF | 17.2 | 13.2 |
| ICCL | 49.5 | 52.8 |
| PPL | 25.0 | 21.0 |
| DKCIL (weak) | 53.4 | 55.5 |
| DKCIL (unlabeled) | 48.7 | 48.6 |
| DKCIL (few-shot) | 51.3 | 52.3 |

**Training with weakly supervised, unlabeled or few-shot datasets.** ❶ The performance predictor of D-$k$-ICL can also be trained with weakly supervised or unlabeled datasets. In earlier experiments, the training split was used as the retrieval set and the test split as the evaluation set. In practice, however, high-quality labeled retrieval sets may not be available. In such cases, weak supervision can be employed. For example, the TREC dataset contains both coarse-grained (6-class) and fine-grained (50-class) labels. Annotating retrieval data with 50 fine-grained labels is costly, whereas coarse 6-class labeling is considerably more efficient. As shown in Section 6, the performance predictor of D-$k$-ICL generalizes across datasets: training with 6-class labels and evaluating on the 50-class dataset still yields the best accuracy of 55.5% (Table 7). Similarly, when no labels are available, we use GPT-4 9B to generate 6-class pseudo-labels. The predictor trained with these pseudo-labels also achieves the best accuracy of 48.7% (Table 7). ❷ The performance predictor of D-$k$-ICL can further be trained in few-shot settings. In previous experiments, 30% of the retrieval dataset was randomly sampled for training. Here, we restrict training data to only 30 examples. As described in Section 4, training requires (`text`, `in-context`, `actual performance`) tuples. When `text` data are limited, additional tuples can be generated by pairing each text with multiple in-context demonstrations from diverse ICL methods. Specifically, we use TTF, DKNN, KATE, CD, ICCL, D-$k$-ICL, and PPL to construct demonstrations for 25 texts, producing 175 (`text`, `in-context`, `actual performance`) tuples. As shown in Table 7, D-$k$-ICL continues to perform strongly under this few-shot setting, reaching a maximum accuracy of 52.3%.

**Cost:** D-$k$-ICL's training on SST-5 takes 18.7 minutes and 468 MB storage, with 0.28s average inference time per text. Although these computational and storage requirements are non-negligible, they represent a worthwhile trade-off given the substantial performance gains achieved by D-$k$-ICL.

# 8 CONCLUSION

We find that dynamically selecting the number of in-context demonstrations based on the query text and the test LLM during inference yields superior performance compared to using a fixed number as a static hyperparameter. To capitalize on this observation, we propose D-$k$-ICL, a method that adaptively determines the number of demonstrations conditioned on both the specific input and the LLM in use. Extensive evaluations show that D-$k$-ICL achieves considerable performance across 3 LLMs, 5 tasks, and 8 datasets, and remains highly effective when applied to the commercial LLM GPT-4o. The method also demonstrates remarkable generalization capability, transferring effectively across different LLMs, datasets, and tasks. Furthermore, D-$k$-ICL functions effectively as a plug-and-play module to enhance existing ICL methods.

ETHICS STATEMENT

This work complies with the ICLR Code of Ethics. The study involved no human subjects or animal experimentation. All datasets, including Emotion, SST5, Gigaword, Gigatiny, STS14, STSB, WNT19-En-Zh, and EN-CS were obtained in accordance with relevant usage guidelines without privacy violations. We implemented measures to prevent biases and discriminatory outcomes throughout the research process. No personally identifiable information was utilized, and no experiments raised privacy or security concerns. We maintain full commitment to research transparency and integrity.

REPRODUCIBILITY STATEMENT

We have implemented comprehensive measures to ensure reproducibility. All code and datasets will be publicly available in an anonymous repository to support verification and replication. The paper details the experimental setup, including training procedures, and model configurations. We employ the publicly available datasets, including Emotion, SST5, Gigaword, Gigatiny, STS14, STSB, WNT19-En-Zh, and EN-CS, ensure consistent and reproducible evaluation outcomes. These resources will enable researchers to validate our work and advance the field.

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

OVERVIEW OF THE APPENDIX

This appendix includes our supplementary materials as follows:

- The LLM usage statement in Section A
- More detailed descriptions of the evaluation metrics in Section B
- Formal definitions of the tasks in Section C
- More details about the datasets in Section D
- Models access information and URLs in Section E

## A  LLM USAGE STATEMENT

Large Language Models (LLMs) assisted in manuscript preparation by enhancing language quality, improving readability, and ensuring textual clarity. The models supported sentence refinement, grammatical correction, and improved narrative flow.

Crucially, LLMs were not involved in conceptual development, research methodology, or experimental design. All intellectual contributions, analytical frameworks, and research concepts originated from the authors. LLM assistance was strictly limited to linguistic enhancement, with no participation in scientific content creation or data analysis.

The authors assume complete responsibility for all manuscript content, including LLM-polished text. We confirm adherence to ethical standards, ensuring no plagiarism or scientific misconduct occurred.

## B  METRICS

This section provides formal definitions and computational methodologies for the four evaluation metrics employed in our study to assess the performance of large language models across diverse tasks.

### B.1  BLEU

The Bilingual Evaluation Understudy (BLEU) metric is predominantly utilized for evaluating the quality of machine-generated text, particularly in machine translation, by comparing it to one or more human-written reference translations. It operates by calculating modified n-gram precision scores, which penalize candidate translations that overgenerate "reasonable" words. The final BLEU score is a weighted geometric mean of individual n-gram precisions up to a specified order $N$ (typically $N = 4$), multiplied by a Brevity Penalty (BP) factor that penalizes candidates shorter than their references.

The computation is formally defined as follows:

$$\text{BLEU} = \text{BP} \cdot \exp\left(\sum_{n=1}^{N} w_n \log p_n\right) \tag{8}$$

where:

- $p_n$ is the modified n-gram precision for n-grams of length $n$, calculated as:

$$p_n = \frac{\sum_{C \in \{\text{Candidates}\}} \sum_{\text{n-gram} \in C} \text{Count}_{\text{clip}}(\text{n-gram})}{\sum_{C' \in \{\text{Candidates}\}} \sum_{\text{n-gram}' \in C'} \text{Count}(\text{n-gram}')} \tag{9}$$

  Here, $\text{Count}_{\text{clip}}$ is the maximum number of times an n-gram appears in any single reference translation, clipped by the count of that n-gram in the candidate translation.
- $w_n$ is the positive weight assigned to each n-gram precision, typically $w_n = 1/N$.

- BP is the Brevity Penalty, which addresses the inherent bias of precision-based metrics against conciseness. It is defined as:

$$BP = \begin{cases} 1 & \text{if } c > r \\ e^{(1-r/c)} & \text{if } c \leq r \end{cases} \tag{10}$$

  where $c$ is the total length of the candidate translation corpus, and $r$ is the effective reference corpus length, typically computed as the sum of the lengths of the closest reference sentences for each candidate.

A higher BLEU score (range 0 to 1, often expressed as a percentage) indicates a stronger alignment between the candidate and reference texts.

## B.2 ROUGE-1

Recall-Oriented Understudy for Gisting Evaluation (ROUGE) is a set of metrics designed for evaluating automatic summarization and text expansion. Unlike the precision-oriented BLEU, the ROUGE-N variant focuses on recall, measuring the proportion of n-grams in the reference summary that are captured by the generated summary. We employ ROUGE-1, which operates on unigrams (single words), to assess the adequacy of content coverage.

The ROUGE-1 recall score is calculated as:

$$\text{ROUGE-1}_{\text{Recall}} = \frac{\sum_{s_r \in S_{\text{ref}}} \sum_{u \in s_r} \text{Count}_{\text{match}}(u)}{\sum_{s_r \in S_{\text{ref}}} \sum_{u \in s_r} \text{Count}(u)} \tag{11}$$

where:

- $S_{\text{ref}}$ is the set of reference summaries.
- $u$ is a unigram.
- $\text{Count}_{\text{match}}(u)$ is the number of times a unigram $u$ appears in both the candidate summary and the reference summaries, clipped by the count in the candidate (for multiple references, the maximum overlap is used).

Often, the F1 score, which is the harmonic mean of unigram precision ($P$) and recall ($R$), is reported to provide a balanced measure:

$$\text{ROUGE-1}_{\text{F1}} = 2 \cdot \frac{P \cdot R}{P + R} \tag{12}$$

A higher ROUGE-1 F1 score signifies better performance in capturing the salient content of the source or reference text.

## B.3 MEAN SQUARED ERROR (MSE)

Mean Squared Error (MSE) is a standard metric for regression tasks, which we utilize for evaluating Semantic Textual Similarity (STS). In STS, models predict a continuous similarity score between two text segments. MSE quantifies the average squared magnitude of the differences between the predicted values ($y_i$) and the actual ground truth values ($\hat{y}_i$). By squaring the errors, MSE disproportionately penalizes larger deviations.

The MSE for a set of $n$ predictions is given by:

$$\text{MSE} = \frac{1}{n} \sum_{i=1}^{n} (y_i - \hat{y}_i)^2 \tag{13}$$

A perfect model would achieve an MSE of 0.0. Consequently, a lower MSE value indicates superior performance, as it reflects a smaller average error in the model's similarity predictions.

## B.4 ACCURACY

Accuracy is a fundamental metric for evaluating performance in classification tasks. It measures the fraction of predictions (both positive and negative) that the model classified correctly out of the total number of instances.

The formula for accuracy is:

$$\text{Accuracy} = \frac{|\text{Correct Predictions}|}{|\text{Total Instances}|} = \frac{\text{TP} + \text{TN}}{\text{TP} + \text{TN} + \text{FP} + \text{FN}} \tag{14}$$

where:

- TP (True Positives) are the positive instances correctly predicted as positive.
- TN (True Negatives) are the negative instances correctly predicted as negative.
- FP (False Positives) are the negative instances incorrectly predicted as positive.
- FN (False Negatives) are the positive instances incorrectly predicted as negative.

Accuracy ranges from 0 to 1 (often expressed as a percentage), where a higher value denotes a greater proportion of correct predictions and thus better model performance.

## C TASK

This section provides formal definitions and contextual background for the five primary Natural Language Processing (NLP) tasks evaluated in this study to demonstrate the capabilities and generalizability of the proposed model.

### C.1 TEXT CLASSIFICATION

Text Classification is a fundamental supervised learning task in NLP that involves assigning a predefined categorical label (or labels) to a given text document based on its content and semantics. Formally, the goal is to learn a mapping function $f : \mathcal{X} \rightarrow \mathcal{Y}$ from an query text space $\mathcal{X}$ to a discrete label space $\mathcal{Y}$. This task is pivotal for applications requiring the organization, structuring, and categorization of textual data, such as sentiment analysis, topic labeling, spam detection, and intent classification. Performance is typically quantified using metrics such as **Accuracy**, which measures the proportion of instances correctly classified over the total number of instances.

### C.2 TEXTUAL SIMILARITY ESTIMATION

Textual Similarity Estimation, often referred to as Semantic Textual Similarity (STS), is a core regression task focused on quantifying the degree of semantic equivalence between two text segments. The objective is to predict a continuous similarity score $s \in [s_{min}, s_{max}]$ that reflects the semantic proximity of a pair of texts $(t_i, t_j)$, moving beyond mere lexical overlap to capture deeper linguistic meaning. This task is critical for applications like information retrieval, duplicate detection, and semantic search. Model performance is rigorously evaluated by measuring the disparity between predicted similarity scores and human-annotated ground truth values, most commonly using the **Mean Squared Error (MSE)**.

### C.3 ABSTRACTIVE SUMMARIZATION

Abstractive Summarization is an advanced text generation task that requires producing a concise and coherent summary $\mathcal{S}$ from a longer source document $\mathcal{D}$, which accurately encapsulates its core semantic content. Unlike *extractive* summarization—which selects and compiles existing phrases or sentences from the source—the abstractive approach involves interpreting the source material, internalizing its meaning, and generating novel phrases and sentences to convey the salient information. This necessitates deep language understanding and generation capabilities. The quality of the generated summaries is conventionally assessed by measuring the lexical or semantic overlap with human-authored reference summaries using metrics such as **ROUGE-N** (e.g., **ROUGE-1**).

## C.4 TEXT EXPANSION

Text Expansion is the task of elaborating a short, potentially underspecified text input (e.g., a set of keywords, a headline, or a telegraphic phrase) into a longer, more detailed, fluent, and coherent text. The model must act as a contextualizing engine, inferring implicit information and generating relevant content that is semantically consistent with the source's intent without introducing hallucinations. This task evaluates a model's ability to perform controlled, knowledge-augmented generation and has practical applications in content creation, writing assistance, and data-to-text systems. Evaluation often involves comparing the system's output to human-authored expansions using n-gram overlap metrics like **ROUGE-1**.

## C.5 MACHINE TRANSLATION

Machine Translation (MT) is the canonical task of automatically translating a text sequence from a source language ($L_s$) into a target language ($L_t$). The principal objective is to learn a conditional mapping $P(y_t|x_s)$ that produces a translation which is not only syntactically well-formed in $L_t$ but also semantically faithful to the source text $x_s$ in $L_s$, preserving its meaning, nuance, and style. It is a profound challenge in NLP, requiring handling of divergent linguistic structures, disambiguation, and cultural specificity. The quality of machine-translated text is automatically evaluated by comparing it to human-produced reference translations using the **BLEU** (Bilingual Evaluation Understudy) metric, which calculates a modified n-gram precision score.

## D DATASET

Table 8: Prompts used for different datasets

| Dataset | Prompt |
|---|---|
| Emotion | You are a helpful assistant. Predict the label of the input text, only give me the label is enough, for instance, label = 'Anger', label = 'Fear', label = 'Joy', label = 'Love', label = 'Sadness', label = 'Surprise'. Labels are Anger, Fear, Joy, Love, Sadness, and Surprise, not other labels. |
| SST5 | You are a helpful assistant. Predict the label of the input text, only give me the label is enough, for instance, label = 'very negative', label = 'negative', label = 'neutral', label = 'positive', label = 'very positive'. |
| SST14 | You are a helpful assistant. You are asked to predict the semantic textual similarity of every input text pairs. Your response only contains a single numerical value with the range from 0 to 5. A larger number indicates a higher degree of similarity. |
| STSB | You are a helpful assistant. You are asked to predict the semantic textual similarity of every input text pairs. Your response only contains a single numerical value with the range from 0 to 5. A larger number indicates a higher degree of similarity. |
| gigatiny | You are a helpful assistant. Expand this paragraph without altering its core meaning. |
| gigaword | You are a helpful assistant. Summarize the following text and generate an abstract. |
| wmt19_Zh-En | You are a helpful assistant. Translate the following text from Chinese to English. |

In this section, we present the datasets utilized in our evaluation, covering a broad spectrum of natural language processing tasks including text classification, semantic similarity, summarization, and machine translation. Each dataset serves as a representative benchmark for its respective task.

## D.1 SST-5

The Stanford Sentiment Treebank (SST-5) is a fine-grained sentiment classification dataset derived from movie reviews. It provides five sentiment labels ranging from "very negative" to "very positive." Each sentence is parsed into a syntactic tree, enabling supervised learning at both the phrase

and sentence levels. SST-5 is widely adopted for benchmarking sentiment analysis models requiring nuanced sentiment discrimination.

### D.2 EMOTION

The Emotion dataset is designed for multi-class emotion classification, containing English sentences annotated with discrete emotional categories such as joy, sadness, anger, fear, surprise, and love. Unlike sentiment classification, which focuses primarily on polarity, this dataset captures diverse affective states, making it suitable for evaluating emotion recognition capabilities in language models.

### D.3 STS14

The Semantic Textual Similarity 2014 (STS14) dataset is part of the SemEval shared task series. It contains sentence pairs annotated with similarity scores ranging from 0 (completely dissimilar) to 5 (semantically equivalent). The dataset covers multiple domains, including newswire, forum discussions, and image captions, thereby serving as a benchmark for semantic similarity estimation.

### D.4 STS15

The Semantic Textual Similarity 2015 (STS15) dataset extends the previous year's benchmark by including more diverse sentence pairs with human-annotated similarity scores. It emphasizes cross-domain generalization and remains a standard testbed for evaluating sentence embedding models on their ability to capture fine-grained semantic relationships.

### D.5 STS16

The Semantic Textual Similarity 2016 (STS16) dataset continues the SemEval STS series with sentence pairs drawn from varied domains, including headlines, answer–answer forums, and question–question forums. It provides gold-standard similarity annotations, thereby enabling evaluation of models' semantic alignment across heterogeneous text sources.

### D.6 STS-B

The Semantic Textual Similarity Benchmark (STS-B) is a consolidated benchmark dataset covering multiple years of the STS shared tasks. It provides human-labeled similarity scores on a continuous scale between 0 and 5. Unlike individual yearly datasets, STS-B offers a standardized benchmark with an official train, development, and test split, facilitating consistent model comparison in semantic similarity research.

### D.7 GIGAWORD

The English Gigaword dataset is a large-scale text corpus consisting of newswire articles from multiple international news agencies. It has been widely used for abstractive summarization tasks, where the goal is to generate a concise headline given a news article sentence or paragraph. Gigaword provides a rich resource for training neural summarization models due to its size and linguistic variety.

### D.8 GIGATINY

Gigatiny is a reduced-scale variant of the Gigaword dataset designed for efficient experimentation in summarization research. By curating a smaller yet representative subset of the original corpus, Gigatiny enables rapid model prototyping and evaluation while retaining the essential characteristics of large-scale summarization tasks.

### D.9 WMT19 EN–ZH

The WMT19 English–Chinese dataset is part of the annual Workshop on Machine Translation (WMT) shared tasks. It provides large-scale parallel corpora for training and evaluating neural

machine translation systems. The dataset covers multiple text domains and reflects real-world translation challenges, making it a primary benchmark for assessing cross-lingual generalization in machine translation systems.

Table 9: Datasets and Their URLs

| Task | Dataset | URL |
| --- | --- | --- |
| Translation | WMT19 En–Zh | `https://huggingface.co/datasets/WillHeld/wmt19-valid-only-zh_en` |
| Textual Similarity Estimation | STS15 | `https://huggingface.co/datasets/mteb/sts15-sts` |
| | STS14 | `https://huggingface.co/datasets/mteb/sts14-sts` |
| | STSB | `https://huggingface.co/datasets/SetFit/stsb` |
| | STS16 | `https://huggingface.co/datasets/mteb/sts16-sts` |
| Abstractive Summarization / Text Expansion | gigatiny | `https://huggingface.co/datasets/SpeedOfMagic/gigaword_tiny` |
| | gigaword | `https://huggingface.co/datasets/Gabriel/gigaword_swe` |
| Text Classification | Emotion | `https://huggingface.co/datasets/dair-ai/emotion` |
| | SST5 | `https://huggingface.co/datasets/SetFit/sst5` |

## E    THE URL OF MODELS

Table 10: Large Language Models and Their URLs

| Model | URL |
| --- | --- |
| GLM4 9B | `https://huggingface.co/zai-org/glm-4-9b` |
| LLAMA-3.1-8b | `https://huggingface.co/meta-llama/Llama-3.1-8B` |
| GPT-4o | `https://platform.openai.com/docs/models/gpt-4o` |
| Qwen2.5 7b | `https://huggingface.co/Qwen/Qwen2.5-7B` |
| LLAMA-3.2-3b | `https://huggingface.co/meta-llama/Llama-3.2-3B` |

