# OpenReview forum: "Dynamic $k$-shot In-Context Learning"
_ICLR.cc/2026/Conference — ICLR 2026 Conference Withdrawn Submission_

### Official Review · Reviewer_1SLG · 2025-10-27

**Soundness:** 3
**Presentation:** 2
**Contribution:** 2
**Rating:** 4
**Confidence:** 4

**Summary:**

This paper proposes a framework for dynamic exemplar allocation in in-context learning (ICL), moving beyond fixed (k)-shot settings. The core component is a neural network that estimates the optimal number of semantically closest exemplars (k) to include for each query, conditioned on the specific LLM in use. The authors demonstrate consistent performance gains across several tasks and datasets and show that the framework generalizes well to other models (e.g., GPT 4o).

**Strengths:**

- Clear empirical motivation: the paper shows that model performance varies non-monotonically with the number of exemplars, supporting the need for a dynamic approach.
- The proposed performance predictor mechanism is conceptually simple and can be easily adapted to other demonstration-based prompting setups.
- Wide selection of benchmarks, accompanied by well-designed ablation studies and generalization analyses.
- The plug-and-play integration and weakly labeled/few-shot training scenarios show practical awareness and an attempt to assess robustness under limited supervision.

**Weaknesses:**

- The paper doesn’t discuss that this approach may struggle in out-of-distribution settings, where finding suitable ICL exemplars is highly challenging.
- The paper does not sufficiently analyze or mitigate the computational overhead introduced by the proposed method. Prior work [1] explicitly addresses such concerns.
- The claim that “little attention has been given to the number of exemplars in ICL” (L80, L124-L129) is inaccurate. Several prior works [2] have investigated this aspect.
- While the main experiments span eight benchmarks, several analyses and follow-up experiments are restricted to at most two datasets, limiting the generality of the claims.
- Numerous citations appear fabricated or unverifiable. For example, all 10 “Ref for” entries (e.g., L617-618, L520-522) refer to nonexistent works despite referencing widely established topics such as AdamW and ROUGE. This raises concerns about citation integrity and reproducibility.

CITATIONS:
- [1] Emily Xiao, Chin-Jou Li, Yilin Zhang, Graham Neubig, and Amanda Bertsch. Efficient Many-Shot In-Context Learning with Dynamic Block-Sparse Attention. In Proceedings of the 63rd Annual Meeting of the Association for Computational Linguistics (Volume 1: Long Papers), pages 31946–31958, Vienna, Austria, July 2025. Association for Computational Linguistics. https://aclanthology.org/2025.acl-long.1542/

- [2] Rishabh Agarwal, Avi Singh, Lei Zhang, Bernd Bohnet, Luis Rosias, Stephanie Chan, Biao Zhang, Ankesh Anand, Zaheer Abbas, Azade Nova, John D. Co-Reyes, Eric Chu, Feryal Behbahani, Aleksandra Faust, and Hugo Larochelle. Many-Shot In-Context Learning. In Advances in Neural Information Processing Systems 37, pages 76930–76966, 2024. https://proceedings.neurips.cc/paper_files/paper/2024/file/8cb564df771e9eacbfe9d72bd46a24a9-Paper-Conference.pdf

**Questions:**

- Could you please provide general statistics on the learned (k) distribution across tasks? As Table 4 suggests, performance depends on the maximum allowed (k).
- Please include a clear explanation (in the appendix) of the baselines, as they directly influence the pool of candidate exemplars in the few-shot training setting.
- The rationale for capping (k) at 10 for comparison is understandable, but it would be informative to explore performance when (k >>> 10).
- Suggested clarifications and minor corrections:
  - Figure 1: expand caption. KATE and CD are not widely known abbreviations.
  - L134: Explain what KATE stands for.
  - L156: reference to a non-existent figure (possibly meant to be Fig. 2).
  - Figure 2: visually appealing but somewhat confusing—numbering the steps would help.
  - L231: remove extra closing parenthesis in (F_{\text{LLM}}).
  - Table 3: for Qwen, training data “WNT19” omits “En–Zh.”
  - L395: “D-k-ICL” is poorly formatted.
  - L423 and L461: likely meant “GLM4 9B.”
  - Table 7: dataset source not mentioned (appears to be TREC).

**Details Of Ethics Concerns:**

No direct ethical issues are apparent beyond the need for accurate citation. However, fabricated or unverifiable references undermine reproducibility and transparency. We found that all (10) citations that have used "Ref for" are hallucinated.

Example: "Ilya Loshchilov and Frank Hutter. Adamw optimizer: Advances and applications in deep learning. Journal of Machine Learning Research, 2025. Ref for ”AdamW optimizerref” in training setup."

---

### Official Review · Reviewer_aNug · 2025-10-28

**Soundness:** 2
**Presentation:** 2
**Contribution:** 2
**Rating:** 2
**Confidence:** 5

**Summary:**

This paper proposes D-k-ICL, which adaptively selects the number of in-context examples for each query using a learned performance predictor. Instead of relying on a fixed k, the method estimates task performance across candidate values and chooses the best one per instance. Experiments on multiple datasets and LLMs show modest but consistent improvements over fixed-k baselines.

**Strengths:**

- Addresses an overlooked yet important factor in ICL: how many examples to use.


- Model- and task-agnostic; can be applied to different LLMs and domains.


- Demonstrates steady gains across several datasets and model scales.

**Weaknesses:**

- The idea of learning a predictor to determine the optimal number of demonstrations feels incremental relative to existing meta-ICL or performance-prediction frameworks. Prior work has explored adaptive retrieval [1,2]. Thus, the novelty lies primarily in applying these ideas to the dimension of “number of shots,” which is conceptually interesting but not deeply transformative.


- Although D-k-ICL demonstrates improved results across multiple datasets and LLMs, the comparisons are limited to heuristic retrieval methods. The baseline pool excludes recent training-based ICL methods such as EPR [3], CEIL [4], and MoD [5], which explicitly optimize retrieval or demonstration selection using supervision. Including these would provide a more comprehensive and fair comparison to assess whether D-k-ICL brings substantive improvements beyond existing learning-based frameworks.


- The scalability of D-k-ICL raises practical concerns. The method requires constructing labeled training data across varying k values, leading to a training cost that grows linearly with k. In contrast, simpler retrieval-based strategies (e.g., top-k selection) can trivially adapt to different k without retraining. The paper does not analyze the computational or annotation overhead relative to these simpler baselines.

**Questions:**

- The paper states that “all code and datasets will be publicly available in an anonymous repository,” but no link or supplementary material is currently provided. Could the authors clarify whether these resources will be shared during the review process to support reproducibility?


- Section 3 motivates the challenge by showing strong performance fluctuations with different k. Does the proposed method reliably find the optimal or near-optimal k for each instance? Quantitative evidence (e.g., distance from the true best-performing k) would help clarify how well the predictor performs the intended selection.


- Line 156 references “Fig. ??,” suggesting a missing figure label. Please verify and correct this cross-reference.


- In Section 3, Naik et al. (2023) are cited as a method for “Cluster-Diversity” within ICL. My understanding is that their work is not originally designed for in-context learning nor explicitly focused on cluster-based diversity. This raises concerns about the accuracy of related-work positioning and the professional of paper writing. I invite the authors to clarify this connection.


- The acronym “SICL” appears in Sections 5.3 and 6 without definition. Is this intended to denote D-k-ICL or another variant? Please clarify the terminology for consistency.


[1]Y Zhang, et al. Active example selection for in-context learning.


[2]Z Chen, et al. MAPLE: Many-Shot Adaptive Pseudo-Labeling for In-Context Learning.


[3]Rubin, et al. Learning to retrieve prompts for in-context learning.


[4]Ye J, et al. Compositional exemplars for in-context learning.


[5]S Wang, et al. Mixture of demonstrations for in-context learning

---

### Official Review · Reviewer_obmn · 2025-11-01

**Soundness:** 2
**Presentation:** 3
**Contribution:** 2
**Rating:** 4
**Confidence:** 4

**Summary:**

This paper empirically finds the inference performance of ICL exhibits considerable variation across datasets and LLMs as the number of demonstrations changes, and proposes to dynamic choose the number of contexts in the ICL. Specifically, this paper constructs the triplet including (text, in-context, actual performance) to train a performance predictor, then use this predictor to predict the best number of contexts during inference.

**Strengths:**

S1: The problem is realistic and important.

S2: The paper is easy to read and easy to follow.

S3: The proposed methods are easy to understand and somewhat reasonable.

S4: The experiments show the effectiveness of the proposed method.

**Weaknesses:**

W1: Why do we not consider the order, the formatting, or other issues in such scenario? For example, only using cosine similarity to determine the order.

W2: When testing, do we need to go through all demonstrations? This will cause a huge complexity and slow the inference speed. In addition, we still have a hyper-parameter, that is, $k$ in equation 6, which means there is still a trade-off, i.e., large $k$ will slow the inference speed but have a reliable guarantee. Besides the “cost” paragraph, can we compare the complexity of the proposed method to the existing fixed-k method?

W3: If the test data is unseen, can the performance predictor generalize to such dataset? In other words, if we randomly split the train / test data from the same dataset, we can train a fixed number of contexts in each dataset.

W4: How does the scalability of this method? In other words, if the model is large (14B, 32B, 70B, etc.), how does the improvement change? Due to the large model may need a few demonstrations to learn. In addition, this paper should categorize baseline methods more clearly, for example, which methods belong to which category summarized in the related work session.

W5: Several presentation issues, such as “Fig. ??” in line 156. In addition, Figure 2 is not clear enough for demonstrating both the big picture of the whole framework and the corresponding details. I suggest providing one more figure for illustration.

**Questions:**

See the weaknesses part for the question.

---

### Official Review · Reviewer_mqhd · 2025-11-03

**Soundness:** 2
**Presentation:** 1
**Contribution:** 2
**Rating:** 2
**Confidence:** 4

**Summary:**

This paper proposes Dynamic-k In-Context Learning (D-k-ICL), a novel approach that adaptively determines the optimal number of demonstrations for each query in in-context learning (ICL), rather than using a fixed hyperparameter k. The authors first conduct an empirical study showing that the optimal number of demonstrations varies significantly across different queries, models, and tasks. Based on this observation, they develop a performance predictor—a neural network that takes both the query text and candidate in-contexts (with varying demonstration counts) as input and estimates the expected inference performance. During inference, the predictor selects the candidate in-context most likely to yield the best output. The method is evaluated across 3 LLMs, 5 tasks, and 8 datasets, showing consistent improvements over baseline methods.

**Strengths:**

1. This paper focuses on an interesting and unexplored ICL problem of determining the number of shots in ICL.

**Weaknesses:**

1. The main concern is that there is a simple baseline to adjust the number of shots in the ICL. That is, introducing the threshold of scores instead of the threshold of top k. This baseline should be included in the comparison.
2. The approach introduces additional computational costs during training and inference, which would prevent this method from the real world applications.
3. The presentation of the work could be further improved.

**Questions:**

1. How sensitive is the performance predictor to the size and quality of training data? What happens with very limited training examples?
2. The performance of ICL also depends on the order of input demonstrations. Can the proposed method take this into consideration?
3. Could simpler heuristics (e.g., using similarity threshold or query complexity metrics to determine k) achieve similar performance gains with lower computational cost?
4. For practical deployment, what is the break-even point where the performance gains justify the additional computational costs?

---

### Note · Authors · 2026-01-06

I have read and agree with the venue's withdrawal policy on behalf of myself and my co-authors.